# Worst-Case Analysis for Randomly Collected Data

**Justin Y. Chen**
MIT
justc@mit.edu

**Gregory Valiant**
Stanford University
gvaliant@cs.stanford.edu

**Paul Valiant**
IAS and Purdue University
pvaliant@gmail.com

## Abstract

We introduce a framework for statistical estimation that leverages knowledge of how samples are collected but makes no distributional assumptions on the data values. Specifically, we consider a population of elements $\{1, \ldots, n\}$ with corresponding values $x_1, \ldots, x_n$. We observe the values for a *sample* set $A \subset \{1, \ldots, n\}$ and wish to estimate some statistic of the values for a *target* set $B \subset \{1, \ldots, n\}$ where $B$ could be the entire set. Crucially, we assume that the sets $A$ and $B$ are drawn according to some known joint distribution $(A, B) \sim P$ over pairs of subsets of $\{1, \ldots, n\}$. A given estimation algorithm is evaluated based on its *worst-case, expected error* where the expectation is with respect to the distribution $P$ from which the sample $A$ and target set $B$ are drawn, and the worst-case is with respect to the data values $x_1, \ldots, x_n$. Within this general framework we give an efficient algorithm to find an estimator for the target mean, as a weighted combination of the input sample—where the weights are a function of the distribution $P$ and the identities of the elements in the sample and target sets $A, B$. We show that the worst-case expected error achieved by this estimator is at most a multiplicative $\pi/2$ factor worse than the optimum for such estimators. A component of this algorithm can also be used to approximate the worst-case expected error of a given estimator. The algorithm and proof leverage a surprising connection to the Grothendieck problem. We extend these results to the setting of linear regression, where each datapoint is not a scalar but a labeled vector $(x_i, y_i) \in \mathbb{R}^{d+1}$. Our framework, which makes no distributional assumptions on the data values but rather relies on knowledge of the data collection process via the distribution $P$, is a significant departure from the typical statistical estimation framework and introduces a uniform algorithmic analysis for the many natural settings where membership in a sample may be correlated with data values, such as when probabilities of sampling vary as in "importance sampling", when individuals are recruited into a sample via a social network as in "snowball sampling" or "respondent-driven sampling" [12, 14] or when samples have chronological structure as in "selective prediction" [10, 21]. We experimentally demonstrate the benefit of this framework and our algorithm in comparison to standard estimators, for several such settings.

## 1  Modeling Data Collection

For many real-world estimation or prediction problems, it is not unreasonable to assume that data values are drawn independently from some underlying distribution. Correspondingly, there is an enormous body of work developing algorithms suited for such settings as well as for related settings with slightly weaker assumptions such as exchangeability or assumptions found in various models in robust statistics or robust learning. By contrast, there are also many settings in which we know very little about the underlying data values, and any sort of distributional assumption would be problematic. For such settings, however, we might have some knowledge or control over the process by which data is collected. How can we design algorithms for estimating basic statistics that are optimal for a

given data collection process? For which data collection processes is accurate estimation or learning possible, *even for worst-case data*? Surprisingly, there seems to be little work on such questions.

We introduce a general framework in which to study these questions. Consider a set of $n$ indices $\{1, ..., n\}$ with corresponding values $x = \{x_1, ..., x_n\}$, and a distribution $P$ over pairs of subsets $(A, B)$ where $A, B \subset \{1, ..., n\}$. We call $A$ the *sample* set and $B$ the *target* set. At inference time, a pair is drawn $(A, B) \sim P$ and we are given access to the data values in $x$ indexed by set $A$, namely $x_A$. Our task is to use $A, B,$ and $x_A$ to estimate some statistic of the values indexed by set B, $\sigma(x_B)$. For example, if $\sigma()$ is the arithmetic mean and $B = \{1, ..., n\}$, then our goal is to use the sample to estimate the population mean.

Given an estimator, we consider the *worst-case, expected error*. The worst-case is with respect to the data values $\{x_i\}$, and the expectation is with respect to the distribution over subsets $P$. This model corresponds to the setting where we have knowledge of the sample collection process, but do not make distributional assumptions on the data values. Throughout, we will be interested in both the question of evaluating the worst-case expected error of a given estimator, as well as the more challenging task of computing the best estimator for a given $P$—namely the estimator that minimizes this worst-case expected error under the sampling distribution $P$.

Below, we illustrate how this framework captures common data collection processes including processes in which certain individuals are biased towards or against being sampled, processes with dependencies such as "snowball sampling" or "chain sampling" where membership in the sample is governed by stochastic processes (e.g. over a social network), and settings where samples must satisfy chronological constraints (e.g. one uses a sample from the past to make predictions about a target in the future). We begin with a standard example.

**Example 1** (Independent Samples). *Consider the setting where the distribution $P$ corresponds to including each element $i \in \{1, \ldots, n\}$ in the sample set $A$ independently with probability $p$, and the goal is to estimate the mean of the target set $B = \{1, \ldots, n\}$. In this setting, provided the data values $x_1, \ldots, x_n$ are bounded (or have bounded variance in the sense that $\frac{1}{n} \sum_i x_i^2 = O(1)$), then the sample mean will concentrate, and even for worst-case datasets, the estimator that simply returns the sample mean will have expected squared error $O(\frac{1}{pn})$. In the related setting where each $i$ is included in the sample independently, but with a possibly distinct probability $p_i$, the worst-case expected error framework corresponds to not making assumptions on how the data values $x_i$ vary with the sampling probabilities $p_i$. An estimator for this setting with small worst-case expected error will have good performance even if the $x_i$'s are, for example, correlated with the $p_i$'s in a pernicious way.*

The example above illustrates one way in which accurate estimation is possible, even for worst-case data: namely if the sample mean (and target mean) concentrates. The following example illustrates that accurate prediction can still be possible, even for a distribution for which the target and sample means have *constant* variance with respect to the randomness in $(A, B) \sim P$. This example captures a setting where the distribution, $P$, respects *chronological* constraints, in the sense that, for any target/sample sets that have non-zero probability under $P$, if $i$ is in the sample set and $j$ is in the target set, $i < j$. Such constraints mirror the many settings where the sample set corresponds to past data, and the target set corresponds to data that will be received in the future. Here, the worst-case component of our framework corresponds to not making assumptions that the world is "stationary"—future elements might not be like past elements.

**Example 2** ("Selective Prediction"). *Consider the joint sample/target distribution $P$ corresponding to the following process: a time $t$ is drawn uniformly from $\{1, \ldots, n-1\}$, and the sample set is $\{1, \ldots, t\}$. Then $w$ is drawn uniformly from $\{1, 2, 4, 8, \ldots, 2^{\log n - 1}\}$ and the target set is $\{t + 1, \ldots, min(t + w, n)\}$. This prediction task corresponds to choosing a day at random, and deciding to make a prediction about the average change in the stock market over the next $w$ days. In this setting, the main results in [10, 21] imply that if the data values are all bounded, then there exists an estimator for the target mean whose worst-case expected squared error is $O(1/\log n)$, and that this is optimal to constant factors. The prediction algorithm achieving this performance is extremely simple: when asked to predict the mean of the next $w$ data items after $t$, return the mean of the most recent $w$ sample values, $\frac{1}{w} \sum_{i=t-w+1}^{t} x_i$. The surprising aspect of this example is that subconstant expected error is achievable, despite there being no distributional assumptions on the values and hence no guarantees that future data are like past data. The randomness in both time $t$ and the window length $w$ of the target set (which define the distribution $P$) are both essential for achieving*

*subconstant worst-case expected accuracy: if either $t$ or $w$ is any fixed value, then the worst-case estimation error becomes constant.*

A third example that fits cleanly within our framework is the class of data collection schemes referred to as *snowball sampling*, *respondent driven sampling* or *chain sampling* [12, 14]. In such a scheme, people who have contributed data are asked (or incentivized) to recruit their acquaintances to contribute data, and the pool of respondents grows, like a snowball. These schemes are frequently used to collect data from sensitive populations, such as drug users. Our framework provides a natural way to design and evaluate estimators for data collected via such sampling processes:

**Example 3** (Snowball Sampling). *Suppose elements $\{1, \dots, n\}$ are located at nodes of a social network. A sample is drawn by independently selecting one (or several) indices; each element in the sample then repeatedly "recruits" each of its friends in the social network (say independently with probability $p$). The sample will then correspond to the elements that have been recruited in the first $t$ iterations of this "viral" process. The target set could correspond to those nodes recruited in iterations $t+1, \dots t+w$ for some horizon $w$, or the target set could be the entire population, $\{1, \dots, n\}$. How do structural properties of the underlying social network translate into positive or negative results on the worst-case expected performance of standard estimators, or an optimal estimator? And how does such an optimal estimator leverage knowledge of the network structure? While we do not have simple rules-of-thumb for these questions, our main results apply to such snowball sampling processes, and we evaluate our algorithm empirically in such a setting in Section 3.*

While the notion of worst-case expected error is in terms of a sampling distribution, $P$, one of the main motivations for this framework is to guide the choice of estimator for the many real-world settings in which the actual sampling distribution is *unknown*. In such settings, one could construct various different plausible $P$'s—for example capturing different hypothetical types of correlation in inclusion in the sample, different response rates for different hypothetical demographic groups, etc. Using the worst-case expected error framework, one could then rigorously evaluate the stability of potential estimators with respect to these various "plausible" $P$'s. This may prove to be a useful alternative to the more standard approach of evaluating estimators with respect to different assumptions on the data values.

## 1.1 Summary of Results

Our main results are efficient algorithms for approximately computing the worst-case expected error of a given estimator with respect to a given distribution $P$ (or sample access to $P$), and for finding an estimator that approximately minimizes this error with respect to $P$. We state these results in the setting where the statistic of interest is the *mean* of the target set. Such results immediately extend to yield analogous statements for any statistic that can be expressed as the average of some functional of each data point (e.g. the variance, or higher moments). We further apply these results in a black-box fashion to yield results in the setting where each datapoint is a labeled vector $(x_i, y_i)$, and the "statistic" of interest is the optimal linear regression model $\beta$ for the target set, so that $y_i \approx x_i^\top \beta$.

Our results in the mean estimation setting focus on *semilinear* schemes, in which estimates can be expressed as a linear combination of the sample values, where the linear coefficients can depend arbitrarily on the indices (but not values) of the sample and target sets $A$, $B$ and the distribution $P$.

**Definition 1.** *A* semilinear *estimation algorithm, $L$, is a mapping from a set of sample elements $A = (\alpha_1, \dots, \alpha_{|A|}) \subset \{1, \dots, n\}$ and set of target elements $B = (\beta_1, \dots, \beta_{|B|}) \subset \{1, \dots, n\}$ to a list of $|A|$ weights, $w_1, \dots, w_{|A|}$ that may depend on $A$, $B$, and $P$. The estimate produced by $L$ when given the sample values $x_A = \{x_{\alpha_1}, \dots, x_{\alpha_{|A|}}\}$ is $\sum_{i=1}^{|A|} w_i x_{\alpha_i}$.*

Intuitively, semilinear algorithms work by mapping the sample/target sets to linear coefficients of the sample values, effectively specifying separate linear estimators for each possible sample/target pair $(A, B)$ supported on $P$. A good semilinear algorithm will produce a set of estimators such that there is no assignment to values $x$ that incurs high error on many of the possible sample/target sets. Such an algorithm will have low worst-case expected error, which is our goal.

Although the class of semilinear algorithms is a restriction, it is a natural starting point for studying this general framework. Additionally, nearly all estimation algorithms that we are aware of fall into this class. The sample mean is trivially semilinear; the more surprising constant-factor optimal scheme in the selective prediction setting of Example 2 produces a semilinear algorithm where the

weights depend on the target indices, not just on the sample indices (recall that the returned estimate is the average of the $w$ highest-indexed sample values, where $w$ is the size of the target set). In the setting of "importance sampling" where each element $i$ is included in the sample independently, with probability $p_i$, standard estimators are also semilinear in that they return a weighted mean of the sample values where the weight of $x_i$ is typically a function of $p_i$. Our notion of semilinear in this setting is even more general, in that the weights given to a sample value $x_i$ can depend arbitrarily on both $p_i$, as well as the other probabilities $\{p_{j\neq i}\}$ and the specific set of sampled indices.

Our main result is that there exists an efficient algorithm which, given sample-access to the joint sample/target distribution $P$, returns a semilinear estimation scheme whose worst-case expected error is within a constant factor of the optimal such scheme for $P$.

**Theorem 1.** *Let $P$ denote a distribution over pairs of subsets $(A, B)$ of $\{1, \ldots, n\}$, and let $\epsilon > 0$ be a fixed error parameter. There is an algorithm $L$ which, given sample-access to $P$ and given sets $A = (\alpha_1, \ldots, \alpha_{|A|})$ and $B = (\beta_1, \ldots, \beta_{|B|})$, takes $poly(n, 1/\epsilon)$ samples from $P$, runs in time $poly(n, 1/\epsilon)$, and returns a list of $|A|$ weights, $w_1^{L(A,B)}, \ldots, w_{|A|}^{L(A,B)}$, with the following guarantee:*

*For any values $x = \{x_1, \ldots, x_n\}$ with $|x_i| \leq 1$ and with high probability over $L$'s samples from $P$, the expected squared difference between the estimate $\sum_{i=1}^{|A|} w_i^{L(A,B)} x_{\alpha_i}$ and the mean of the target set $\frac{1}{|B|} \sum_{i=1}^{|B|} x_{\beta_i}$ is within an additive $\epsilon$ and multiplicative $\pi/2$ factor of the worst-case expected error of the optimal semilinear algorithm. Formally,*

$$
\mathop{\mathbb{E}}_{(A,B)\sim P}\left[ \left( \sum_{i=1}^{|A|} w_i^{L(A,B)} x_{\alpha_i} - \frac{1}{|B|}\sum_{i=1}^{|B|} x_{\beta_i} \right)^2 \right]
$$

$$
\leq \epsilon + \frac{\pi}{2}\left( \inf_{L':(A,B)\to\{w_i^{L'(A,B)}\}} \; \sup_{(x'_1,\ldots,x'_n):|x'_i|\leq 1} \; \mathop{\mathbb{E}}_{(A,B)\sim P}\left[ \left( \sum_{i=1}^{|A|} w_i^{L'(A,B)} x'_{\alpha_i} - \frac{1}{|B|}\sum_{i=1}^{|B|} x'_{\beta_i} \right)^2 \right] \right).
$$

As a component of our proof of this theorem, we give an efficient algorithm which approximates the worst-case expected error of any semilinear algorithm to within this multiplicative $\pi/2$ factor (see Proposition 1 for a formal statement). This approximation factor of $\frac{\pi}{2}$ is optimal in the sense that there exist a semilinear estimator and distribution $P$ for which estimating the worst-case expected error to within a $\frac{\pi}{2}$ factor is NP-hard. We discuss this in Section 2.1.

While we focus on semilinear schemes, these are not necessarily optimal (see Appendix D for details):

**Fact 1.** *There exists a distribution $P$ for which the optimal semilinear mean estimation scheme achieves a worst-case expected squared error that is larger than the worst-case expected squared error of the optimal (unconstrained) scheme.*

As we discuss in Section 1.3, one open question is to understand the severity of this gap between semilinear versus arbitrary estimation algorithms. We conjecture this gap is bounded by a small constant; the most extreme gap that we know of, found via an automated search, is a factor of 1.004.

**Linear Regression.** We also consider the following natural extension of our results to the setting of $d$-dimensional linear regression. As above, there is a joint distribution $P$ over sample/target subsets $A, B \subset \{1, \ldots, n\}$. Each index $i$ has an associated labeled vector, $(x_i, y_i)$ with $x_i \in \mathbb{R}^d$ and $y_i \in \mathbb{R}$. For a sample/target pair $A, B$, we observe the data points $(x_i, y_i)$ for $i \in A$, and the task is to return a linear regression model $\hat{\beta}$ that has small error when applied to the (unobserved) datapoints indexed by the target set $B$. Specifically, letting $\beta_B$ denote the least-squares regression model for the target set $\{(x_i, y_i)\}_{i\in B}$, the goal is to return $\hat{\beta} \approx \beta_B$.

The following theorem leverages Theorem 1; the algorithm and proof are given in Appendix E.

**Theorem 2.** *Consider the regression setting described above corresponding to a sample/target distribution $P$, with the additional guarantee that each coordinate of the features $x_i$ and labels $y_i$ are scaled so as to have magnitude at most 1. Let $\alpha_P$ be the mean squared error guaranteed by Theorem 1 for the (scalar) mean estimation setting. There is a polynomial-time algorithm which, given $A, B$, the datapoints $\{(x_i, y_i)\}_{i\in A}$ and sample-access to $P$, with probability $1 - \delta$ returns*

*regression coefficients $\hat{\beta}$ such that $\|\hat{\beta} - \beta_B\| \leq 3\frac{\sqrt{\alpha_P d^3/\delta}}{\sigma_d^2}$, for any $\delta > 0$; here $\beta_B$ is the true vector of least-squares coefficients for $\{(x_i, y_i)\}_{i\in B}$, and $\sigma_d$ denotes the smallest singular value of the covariance $\frac{1}{|B|}\sum_{i\in B} x_i x_i^\top$. In the case that the features $\{x_i\}_{i\in B}$ are known for the target set (and only the $y_i$'s are unknown) then we have the improved bound that, except with $\delta$ probability, $\|\hat{\beta} - \beta_B\| \leq \frac{\sqrt{\alpha_P d/\delta}}{\sigma_d}$.*

## 1.2 Related Work

There has been significant recent effort developing algorithms for estimation and learning with strong performance guarantees beyond the idealized setting of data drawn i.i.d. from a fixed distribution. This includes the recent body of work on *robust* learning and statistics. Building off a long line of work from the statistics community (see e.g. [15, 24]), the models considered in these works assume that datapoints are drawn independently from some distribution of interest, and then an $\alpha$ fraction of datapoints are corrupted arbitrarily/adversarially. (Some of these works also consider the slightly weaker *contamination* model where the $\alpha$ fraction of arbitrary data is specified before the $1 - \alpha$ fraction of i.i.d. data is drawn.) Recent work has developed computationally efficient algorithms for basic estimation and learning tasks in these settings, beginning with estimating the mean and covariance of a high-dimensional Gaussian [8, 19], and subsequently considering more general optimization problems over data, including linear regression [4, 23, 18, 16]. While this line of work relaxes the typical assumption that *all* datapoints are drawn i.i.d. from a distribution, these works still rely on the assumption that a significant fraction of the data is drawn from a well-behaved distribution. From a technical perspective, these works typically proceed by analyzing the structure of the $1 - \alpha$ fraction of i.i.d. datapoints, and then showing that the adversarial datapoints cannot completely obscure this structure. In this sense, the distributional assumptions on the $1 - \alpha$ fraction of "good" data are critically leveraged.

There is also a line of recent work developing algorithms that work on *truncated* data [6, 7] which captures one commonly arising class of dataset that deviates from the i.i.d. setting. Here, the assumption is that data is drawn independently from a "nice" distribution—a high-dimensional Gaussian in the case of [6]—but then the dataset is truncated, revealing only the portion that lies within some specified set. The challenge is that this conditioning often significantly skews the statistics of the data. Work on learning from truncated samples differs significantly from the framework considered in our work, in that the positive results in [6, 7] leverage assumed structure of the underlying data: the Gaussian assumption in [6], and for [7] the assumptions of an underlying noisy linear model and that the truncation procedure is only a function of the label of each datapoint.

Beyond the dependencies that truncation introduces, recent work also considers regression in a setting with more complex dependencies, that models the type of dependence that may arise when datapoints correspond to nodes within a network [5]. In that work, the authors revisit the standard noisy linear regression model with labels $y_i = \theta^T x_i + \epsilon_i$, and the standard logistic regression model where $Pr[y_i = 1] = 1/(1 + exp(\theta^T x_i))$. Instead of assuming that the $\epsilon_i$ and logistic outcomes are drawn independently, they consider the case where these are generated in a correlated fashion, corresponding to a known/fixed covariance matrix with an unknown strength parameter. Despite these dependencies, the authors provide an efficient algorithm for learning the model, $\theta$, in this setting, that still achieves the error guarantees of the independent settings, provided some mild assumptions are satisfied.

Finally, it is worth clarifying the distinction between our framework and the *on-line* learning framework. As with our framework, much of the work in on-line learning makes no assumptions about the underlying data, and often assumes that the underlying data is adaptively responding to our predictions. Beyond this, however, the frameworks are quite different: our framework considers the task of making a single prediction, as opposed to a sequence of predictions. Additionally, we are measuring the performance of algorithms against all algorithms in their class, instead of in comparison to some set of fixed benchmarks.

## 1.3 Discussion and Open Directions

We introduce a framework for understanding statistical estimation where we make no assumptions on the data values themselves, but model the process $P$ by which sample and target datasets are collected. Within this framework, an estimator is evaluated based on its worst-case expected error,

where the worst-case is with respect to the data values, and the expectation is with respect to the selection of the sample and target sets according to $P$. We present algorithms for approximating the worst-case expected error of an estimator with respect to $P$, and for computing an estimator that approximately minimizes this error, within a broad class of estimators. In addition to the strong theoretical guarantees, this algorithm yields estimators that seems to perform extremely well on several natural synthetic settings where samples are drawn from nontrivial sampling distributions.

There are a number of natural open directions prompted by this work. Can the algorithms be adapted to have a runtime independent of $n$? The framework and definition of worst-case expected error certainly applies to the case where the underlying domain is infinite (instead of $\{1, \ldots, n\}$)—can efficient algorithms be developed for that setting? If data values are constrained to an $\ell_1$ or $\ell_2$ ball (as opposed to the $\ell_\infty$ ball corresponding to our assumption that each value is bounded), are simpler algorithms possible? What is the gap between the worst-case expected error of the best semilinear mean estimation algorithm and the best unconstrained scheme? It also seems worth considering specific classes of distributions from the perspective offered by our framework. For example, for $P$ corresponding to snowball sampling over a social network, what network properties imply subconstant estimation error? Are there variants of snowball sampling that yield significantly better or worse values of the expected estimation error for worst-case data?

Finally, it seems worthwhile extracting high-level interpretable properties of $P$ that imply the existence of estimators with subconstant (or inverse polynomial) error, or properties that imply that the worst-case expected error will be constant for any estimator. For the many cases where we have control over how data is collected, such properties could serve to guide the design of these data collection pipelines.

## 2 Algorithms and Connection to the Grothendieck Problem

Given elements $\{1, \ldots, n\}$ from which both the sample and target sets are drawn, the main component of our model is a joint distribution $P$ on the sample and target sets $(A, B)$. Such a joint distribution can be approximated to arbitrary accuracy as an unweighted distribution over a list of pairs $(A_i, B_i)$, and for ease of notation, we adopt this representation here.

**Definition 2.** *A joint sample/target distribution over a universe of $n$ elements is specified by a list $P$ of pairs $(A_i, B_i)$ of some length $m$, where for each $i \in \{1, \ldots, m\}$ the sets $A_i, B_i$ are subsets of $\{1, \ldots, n\}$. To sample from this distribution, choose a uniformly random $i \sim \{1, \ldots, m\}$ and let $A_i$ be the sample indices, and $B_i$ be the target indices. For values $\{x_1, \ldots, x_n\}$, the sample values will be $x_{A_i}$ and the target values will be $x_{B_i}$.*

Given a sample set $A$ and target set $B$ from such a distribution $P$, for values $x_1, \ldots, x_n$ the algorithmic challenge is to predict a desired attribute of the target values $x_B$, using knowledge of the sample values $x_A$ along with the indices $A$ and $B$. We first focus on the case where the values are real numbers and the goal is to compute the arithmetic mean of the target values. Our objective is to minimize the root-mean-squared error of the estimate of the mean, relative to the scale of the data, $\max_i |x_i|$. Since semilinear estimators scale linearly with the data, this is equivalent to normalizing the data by dividing through by $max_i|x_i|$, and then minimizing the mean squared error subject to the data bound $|x_i| \le 1$, which is the approach we adopt throughout.

**Definition 3.** *For an estimation algorithm $f(x_A, A, B)$ taking as inputs the sample values along with the indices of the sample and target sets, we define the worst-case expected performance for distribution $P$, to be the maximum over values $x_1, \ldots, x_n$ of the mean squared error of its estimate:*

$$\max_{x_1, \ldots, x_n \in [-1,1]} \frac{1}{m} \sum_{i=1}^{m} \left( f(x_{A_i}, A_i, B_i) - mean(x_{B_i}) \right)^2 .$$

For *semilinear* estimators the following notation will simplify the analysis.

**Definition 4.** *Given a sample/target distribution $P = (A_1, B_1), \ldots, (A_m, B_m)$, a semilinear estimation algorithm consists of a vector $a_i \in \mathbb{R}^n$ for each $i \in \{1, \ldots, m\}$, where the support of $a_i$ is a subset of $A_i$. Thus the estimate $f(x_{A_i}, A_i, B_i)$ is simply evaluated as the vector-vector product $a_i^T x$.*

*Correspondingly, we reexpress $mean(x_{B_i}) = b_i^T x$ by defining for each $B_i$ a corresponding vector $b_i \in \mathbb{R}^n$, where $b_i(j) = \frac{1}{|B_i|}$ for $j \in B_i$ and 0 otherwise. Thus the performance of a semilinear*

*estimation algorithm* $(a_i)$ *equals*

$$\max_{x_1,\ldots,x_n \in [-1,1]} \frac{1}{m} \sum_{i=1}^{m} \left( (a_i - b_i)^T x \right)^2 = \max_{x_1,\ldots,x_n \in \{-1,1\}} x^T \left( \frac{1}{m} \sum_{i=1}^{m} (a_i - b_i)^T (a_i - b_i) \right) x. \quad (1)$$

In the second expression above, we (equivalently) restrict the range of each $x_j$ to the endpoints $\{-1, 1\}$ since the expression being maximized is a positive semidefinite quadratic form of $x$, and thus each $x_j$ may be moved to one of the endpoints of its range without decreasing the objective function.

**Definition 5.** *Given a sample/target distribution* $P = (A_1, B_1), \ldots, (A_m, B_m)$*, the performance of the best semilinear estimator is*

$$\frac{1}{m} \min_{a_i : \{j : a_i(j) \neq 0\} \subseteq A_i} \max_{x_1,\ldots,x_n \in \{-1,1\}} \sum_{i=1}^{m} \left( (a_i - b_i)^T x \right)^2 .$$

## 2.1 Worst-Case Performance of a Fixed Semilinear Estimator

Here we consider the challenge of optimizing Equation 1: given a fixed semilinear estimator, how good is it? As noted above, the matrix in the parentheses in the second expression of Equation 1 is positive semidefinite. In fact, for appropriate coefficients $a_i$, we can make the matrix $\frac{1}{m} \sum_{i=1}^{m} (a_i - b_i)^T (a_i - b_i)$ be an arbitrary positive semidefinite matrix (though possibly at the cost of an "unnatural" estimation algorithm). Thus the problem of computing or estimating the performance of a given estimation algorithm is identical to what is known as the positive semidefinite Grothendieck problem. Since the Grothendieck problem includes MAX-CUT, which is NP-hard, evaluating the performance of a fixed estimator is also NP-hard. Further, as was recently shown by Briët, Regev, and Saket, it is NP-hard even to approximate the semidefinite Grothendieck problem to within a factor of $\frac{\pi}{2}$ [17, 3]. Thus, even for a fixed semilinear estimator, we cannot hope to approximate its performance—given by Equation 1—to better than a factor of $\frac{\pi}{2}$ in polynomial time.

However, analogously to the Goemans-Williamson semidefinite relaxation of MAX-CUT, we consider the semidefinite relaxation of the semidefinite Grothendieck problem, replacing each scalar variable $x_j$ with a vector $v_j$ in the $n$-dimensional unit ball. The proof of the following proposition is given in a self-contained fashion in Appendix A, and is based on the analysis of the randomized rounding scheme from Nesterov [20]. In this appendix we also provide some additional explanation and background on the Grothendieck problem and relaxation.

**Proposition 1.** *Given a sample/target distribution* $P = (A_1, B_1), \ldots, (A_m, B_m)$*, the problem of evaluating the performance* $p$ *of a semilinear estimator, specified by vectors* $a_1, \ldots, a_m \in \mathbb{R}^n$*, is NP-hard to estimate to within a multiplicative factor of* $\frac{\pi}{2}$*. However, letting* $M = \frac{1}{m} \sum_{i=1}^{m} (a_i - b_i)(a_i - b_i)^T$*, the optimum of the convex (semidefinite) program*

$$\max_{V \text{ psd}, V_{(j,j)} \leq 1} \sum_{j,k=1}^{n} M_{(j,k)} V_{(j,k)} \quad (2)$$

*is in the interval* $[p, \frac{\pi}{2} p]$*, and can be found in polynomial time by semidefinite programming.*

## 2.2 Computing a Near-Optimal Semilinear Estimator

While Section 2.1 analyzed the problem of evaluating the performance of a fixed semilinear estimator, here instead we aim to find a near-optimal semilinear estimator. This is a challenging setting for optimization, as even evaluating the objective function, to within a factor of $\frac{\pi}{2}$, is NP-hard (as discussed in Section 2.1). However, as we will see, the convex (semidefinite) relaxation derived in Section 2.1 not only lets us approximate the performance of a fixed estimator to a $\frac{\pi}{2}$ factor, but also provides the crucial structure enabling us to find a semilinear estimator whose performance is within a $\frac{\pi}{2}$ factor of the best possible semilinear estimator.

**Theorem 3.** *Algorithm 1, given a description of the joint distribution of sample and target sets* $(A_1, b_1), \ldots, (A_m, b_m)$*, runs in polynomial time, and returns coefficients for a semilinear estimator whose expected squared error is within a* $\frac{\pi}{2}$ *factor of that of the best semilinear estimator. The value of the objective function achieved by* $\widehat{V}$ *is* $m$ *times the Proposition 1 SDP bound on the mean squared error of the best semilinear estimator.*

**Algorithm 1** SDP Algorithm yielding $\frac{\pi}{2}$-approximation to the best semilinear estimator

---

**Input:** A joint distribution $P$ of sample and target sets, expressed as a list of pairs $(A_1, b_1), \ldots, (A_m, b_m)$, where each $A_i \subset \{1, \ldots, n\}$ is the indices of the sample set in the $i^{\text{th}}$ pair, and each $b_i$ is a vector with uniform values over the target set in the $i^{\text{th}}$ pair, as in Definition 4.

For an $n \times n$ matrix $V$ and a set $A_i \in \{1, \ldots, n\}$, let $V_{A_i}$ denote $V$ restricted to the rows in $A_i$, and let $V_{A_i, A_i}$ denote $V$ restricted to *both* rows and columns in $A_i$.

1. Compute the concave maximization

$$\widehat{V} = \underset{V \text{ psd}, V_{j,j} \leq 1}{\arg\max} \sum_{i=1}^{m} b_i^T (V - V_{(A_i)}^T V_{(A_i, A_i)}^{-1} V_{(A_i)}) b_i \qquad (3)$$

2. For each $i \in \{1, \ldots, n\}$, let $a_i$ when restricted to the coordinates $A_i$ equal $\widehat{V}_{(A_i, A_i)}^{-1} \widehat{V}_{(A_i)} b_i$, and 0 on the remaining coordinates. **Output** $a_1, \ldots, a_m$.

---

The function inside the sum in Step 1 can be reexpressed as a standard semidefinite program, using the Schur complement to reexpress the matrix inverse—which is done automatically, for example, in the CVXPY package, as used in our code for the experiments in Section 3. The inverse in Step 2 to compute the linear coefficients can be interpreted as a pseudoinverse $\widehat{V}_{(A_i, A_i)}^{+}$ in cases where it would otherwise be singular.

The proof of Theorem 3 is given in Appendix B. While Algorithm 1 takes as input the entire description of the joint sample/target distribution, such a description might be (1) unavailable in practice and/or (2) have support size $m$ that is exponentially large. In Appendix C we modify this algorithm to use only a polynomial number of *samples* from distribution $P$, addressing both issues mentioned above, establishing Theorem 1.

## 3 Empirical Evaluation

We illustrate the empirical performance of Algorithm 1 for estimating the target mean in three natural settings in which membership in the sample may be perniciously correlated with the underlying data values. The SDP formulation for estimating the worst-case expected squared error of a fixed semilinear estimator in Proposition 1 allows us to compare our algorithm with natural baseline estimators. These experiments are based on our implementation of Algorithm 1 using the Python CVXPY package [9, 1] with the MOSEK solver [2]—our code is available at `https://github.com/justc2/worst-case-randomly-collected`.

**Importance Sampling:** We consider a set of $n = 50$ elements where the $i$th element is included in the sample set independently with probability $p_i$, with $p_1, \ldots, p_{25} = 0.1$ and $p_{26}, \ldots, p_{50} = 0.5$. The target set is the entire population, i.e. the goal is to estimate the population mean. Table 1 compares the performance of the estimator given by our framework with two standard baselines: *inverse proportional reweighting* and *subgroup estimation*. For inverse proportional reweighting, the estimate is given by the weighted mean of the values in the sample set where the weight for value $x_i$ is $1/p_i$. For subgroup estimation, the estimator separately computes the sample mean for $i \leq 25$ and $i > 25$ and then returns the average of these two values. We empirically evaluate the above estimators in terms of expected squared error on the following data values, which are designed to illustrate the strengths and weaknesses of the above estimators. For *constant* values, all values are 1; for *intergroup variance*, $x_1, \ldots, x_{25} = 1$ and $x_{26}, \ldots, x_{50} = -1$; for *intragroup variance*, $x_{even} = 1$ and $x_{odd} = -1$; and for *worst-case*, the approximate worst case error is given by solving the SDP relaxation described in Section 2.1 (note that in this setting the approximation error is negligible).

Even in this simple setting with independent sampling, our algorithm ("SDP Alg") gives roughly a two-fold improvement over the baselines in terms of worst-case expected error. While both inverse proportional reweighting and subgroup estimation are unbiased, they have high variance in some of the settings. As inverse proportional reweighting is non-adaptive to the sample size (it always assigns the same weights to the elements), the size of the sample set has a large impact on bias of the estimate. Subgroup estimation is adaptive, but if we only get a few samples from one of the groups, it

| Data Values | Inv. Prop. Reweighting | Subgroup Estimation | SDP Alg |
|---|---|---|---|
| Constant ($x_i = 1$) | 0.100 | 0.018 | 0.051 |
| Intergroup Variance | 0.100 | 0.018 | 0.053 |
| Intragroup Variance | 0.100 | 0.121 | 0.052 |
| Worst-Case Bound (via SDP) | 0.101 | 0.122 | 0.053 |

**Table 1:** Importance Sampling Experiment: Comparison of expected squared errors of our approach (SDP Alg) and two standard unbiased estimators, across three different assignments to underlying data, together with worst-case bounds given by our SDP. See text for details of the setting.

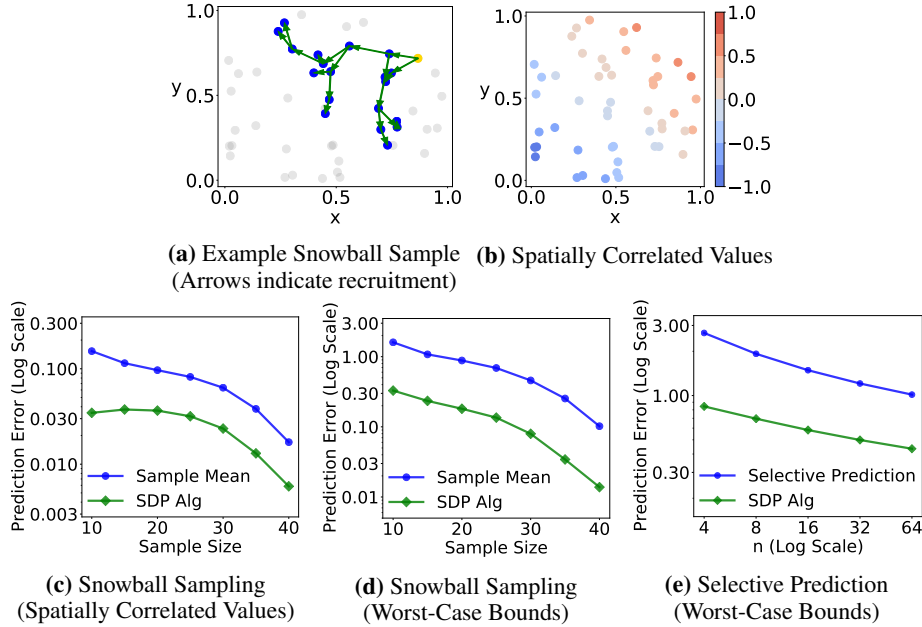

**(a)** Example Snowball Sample (Arrows indicate recruitment)    **(b)** Spatially Correlated Values

**(c)** Snowball Sampling (Spatially Correlated Values)    **(d)** Snowball Sampling (Worst-Case Bounds)    **(e)** Selective Prediction (Worst-Case Bounds)

assigns high weights to those samples, which contribute a high variance. SDP Alg is adaptive to the sampling process, keeping the error small in all cases, by effectively regularizing its estimate to avoid over-reliance on any small number of elements.

**Snowball/Chain Sampling:** In this experiment, an underlying set of $n = 50$ points is drawn uniformly from the 2D unit square. A sample is generated by first including a randomly selected point; then, iteratively, each point in the sample "recruits" two of its five closest neighbors, until the desired sample size is reached. The upper left pane of the figure depicts this process. The target set is the entire population of $n$ individuals. We compare the average squared error of our mean estimation algorithm (labeled SDP Alg) versus that of the naive estimator that returns the sample mean. We consider two settings: 1) the true values are spatially-correlated and are given by the sum of the x and y coordinates of the point (bottom left pane), and 2) the worst-case values for this sampling distribution, approximated by the SDP relaxation from Section 2.1 (bottom middle pane). In both cases, our algorithm outperforms the baseline estimator: by 2–4× for spatially-correlated values (even though our algorithm was not optimizing for this case!) and by 4–7× for worst-case values.

**Selective Prediction:** The selective prediction setting, described in Example 2 and considered in [10, 21], corresponds to chronological prediction problems in which samples from the past are used to make estimates about data in the future. These previous works show that the estimator which, to predict the mean of a target set of size $w$ outputs the mean of the most "recent" $w$ points in the sample, achieves worst-case expected squared error $O(\frac{1}{\log n})$, where $n$ is the total number of elements, and that this is asymptotically optimal. The bottom right pane of the figure compares that estimator with our algorithm, illustrating that both approaches scale with $O(\frac{1}{\log n})$ but that our algorithm yields a 2–3× reduction in error.

## Broader Impact

The question of how to extract accurate statistics based on nonuniform/biased samples is of utmost societal importance. And this basic question is still far from solved—one need only look to the consistent errors across political polls, or more recent discussion on estimating the rate of COVID exposure based on different strategies for recruiting participants and then "correcting" for these biased samples. The vast majority of work on accurate estimation is based on strong distributional assumptions on the data values. The risk is that when these assumptions do not hold, the estimates and their confidence bounds, are meaningless. In this work, we introduce a very general framework that allows one to ask (and answer) the question of whether a given data collection procedure can admit an estimation algorithm which will be accurate, even for worst-case data values. We hope that this framework, which we refer to as *worst case analysis for randomly collected data*, will offer better estimators in some settings, and offer new perspectives on collecting and inferring information from data samples.

## Acknowledgments and Disclosure of Funding

Justin Chen is partially supported by the NSF Graduate Research Fellowship under Grant No. 1745302, the NSF CCF-2006806 award, and the Simons Investigator Award. Gregory Valiant is partially supported by NSF awards AF-1813049, CCF-1704417, and 1804222, ONR Young Investigator Award N00014-18-1-2295, and DOE Award DE-SC0019205. Paul Valiant is partially supported by NSF awards IIS-1562657, DMS-1926686, and indirectly supported by NSF award CCF-1900460.

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
