[Supplementary Material]

# A Proof of Proposition 1

In this section we give a self-contained proof of Proposition 1, restated here for convenience:

**Proposition 1** *Given a sample/target distribution $P = (A_1, B_1), \ldots, (A_m, B_m)$, the problem of evaluating the performance $p$ of a semilinear estimation algorithm specified by vectors $a_1, \ldots, a_m \in \mathbb{R}^n$ is NP-hard to estimate to within a multiplicative factor of $\frac{\pi}{2}$. However, letting $M = \frac{1}{m} \sum_{i=1}^{m} (a_i - b_i)^T (a_i - b_i)$, the optimum of the convex (semidefinite) program*

$$\max_{V \text{ psd, } V_{(j,j)} \leq 1} \sum_{j,k=1}^{n} M_{(j,k)} V_{(j,k)}$$

*is in the interval $[p, \frac{\pi}{2}p]$, and can be found in polynomial time by standard semidefinite programming algorithms.*

The proof of this proposition leverages the connection to the positive semidefinite Grothendieck problem:

**Definition 6.** *The positive semidefinite Grothendieck problem, given an $n \times n$ positive semidefinite matrix $M$ is to evaluate:*

$$\max_{x_1, \ldots, x_n \in \{-1, 1\}} x^T M x \tag{4}$$

*(Note that this problem is sometimes phrased as the optimization over a* pair *of vectors $x, y$, of the expression $x^T M y$, though for positive semidefinite $M$, an optimum will always be attained when $x = y$.)*

The positive semidefinite Grothendieck problem includes MAX-CUT as a special case, since, for an undirected graph $G$, its Laplacian $L$ is positive semidefinite, and for any vector $x \in \{-1, 1\}^n$ that labels its vertices, the value of $x^T L x$ will equal the total degree of the graph plus the size of the cut induced by the labels of $x$. Thus, since MAX-CUT is NP-hard, evaluating the performance of a fixed estimator is also NP-hard. Further, Håstad showed that it is NP-hard to even approximate MAX-CUT to within a multiplicative factor of $\frac{17}{16}$ [13]. For the more general case of the semidefinite Grothendieck problem considered here, Khot and Naor showed the unique-games hardness of approximating the optimum to within a factor of $\frac{\pi}{2}$; this result was recently strengthened by Briët, Regev, and Saket to show it is in fact NP-hard to get an approximation ratio better than $\frac{\pi}{2}$ [17, 3]. Thus, even for a fixed semilinear estimation algorithm, we cannot hope to approximate its performance—given by Equation 1—to within a factor of $\frac{\pi}{2}$.

Analogously to the Goemans-Williamson semidefinite relaxation of MAX-CUT, we consider the semidefinite relaxation of the semidefinite Grothendieck problem, replacing each scalar variable $x_j$ with a vector $v_j$ in the $n$-dimensional unit ball.

**Definition 7.** *Given an $n \times n$ positive semidefinite matrix $M$, the semidefinite relaxation of the positive semidefinite Grothendieck problem is to evaluate:*

$$\max_{v_j \in \mathbb{R}^n : ||v_j|| \leq 1} \sum_{j,k=1}^{n} M_{(j,k)} (v_j^T v_k) \tag{5}$$

*or, equivalently, letting "psd" denote the property of a matrix being positive semidefinite,*

$$\max_{V \text{ psd, } V_{(j,j)} \leq 1} \sum_{j,k=1}^{n} M_{(j,k)} V_{(j,k)}$$

Crucially, the set of positive semidefinite matrices is convex, so thus the optimization problem of Definition 7 (in its second form) maximizes a linear function over a convex set, and thus can be computed in polynomial time.

Goemans and Williamson famously showed, via a randomized rounding scheme, that the gap between MAX-CUT and the result of the induced positive semidefinite relaxation is bounded by a factor of 1.14 [11]. For the more general setting here, of arbitrary positive semidefinite matrices instead of graph Laplacians, Nesterov showed a bound of $\frac{\pi}{2}$ [20]. We include a self-contained derivation here, for the sake of completeness.

Since scaling a single vector $v_j$ affects Equation 5 in a convex quadratic manner, there will always be an optimum of Equation 5 where $||v_j|| = 1$ for all $j$. We assume this, for simplicity, when describing the randomized rounding procedure below.

**Definition 8.** *Given $n$ unit vectors $v_j \in \mathbb{R}^n$, for $j \in \{1, \ldots, n\}$, the Goemans-Williamson randomized rounding procedure chooses a random direction $r$, and for each vector $v_j$ returns a scalar $x_j = \text{sign}(r^T v_j)$.*

**Proposition 2.** *Given an $n \times n$ positive semidefinite matrix $M$, and $n$ unit vectors $v_1, \ldots, v_n \in \mathbb{R}^n$, the value of the relaxed Grothendieck problem, $\sum_{j,k=1}^{n} M_{(j,k)}(v_j^T v_k)$ is at most $\frac{\pi}{2}$ times the expected value of the original Grothendieck problem evaluated on scalars $x_1, \ldots, x_n \in [-1, 1]$ obtained from $v_1, \ldots, v_n$ by the Goemans-Williamson randomized rounding procedure, $\mathbb{E}[\sum_{j,k=1}^{n} M_{(j,k)} x_j x_k]$.*

Thus for any objective value that can be achieved in the relaxed problem, with vectors $v_1, \ldots, v_n$, the original problem can achieve an objective value at least a $\frac{2}{\pi}$ fraction of it, since it does so in expectation over scalars $x_1, \ldots, x_n$ obtained by the randomized rounding procedure.

*Proof of Proposition 2.* As in the analysis of the Goemans-Williamson randomized rounding scheme for MAX-CUT, note that the expected value $\mathbb{E}[x_j x_k] = \mathbb{E}_r[\text{sign}(r^T v_j)\text{sign}(r^T v_k)]$, where $r$ is a randomly chosen direction. Because of the rotational symmetry of the distribution of $r$, we may equivalently rotate $v_j$ and $v_k$ into the plane, from which we can see that, for $r$ also projected into the plane, $\text{sign}(r^T v_j)\text{sign}(r^T v_k)$ equals 1 when $r$ is within $\frac{\pi}{2}$ radians of *both* $v_j, v_k$ or *neither* of them. For a randomly chosen $r$ in the plane, this happens with probability $1 - \frac{1}{\pi}\theta_{j,k}$, where $\theta_{j,k}$ is the angle between $v_j, v_k$, yielding that $\mathbb{E}[x_j x_k] = 1 - \frac{2}{\pi}\theta_{j,k}$.

As $\theta_{j,k}$ may be computed as $\arccos(v_j^T v_k)$, we may express the expected objective value after randomized rounding as

$$\mathbb{E}[\sum_{j,k=1}^{n} M_{(j,k)} x_j x_k] = \sum_{j,k=1}^{n} M_{(j,k)}(1 - \frac{2}{\pi}\arccos(v_j^T v_k))$$

Recall our overall aim, to show that this value times $\frac{\pi}{2}$ is greater than or equal to $\sum_{j,k=1}^{n} M_{(j,k)}(v_j^T v_k)$. Subtracting these two quantities means that we need to show that the following quantity is nonpositive:

$$\sum_{j,k=1}^{n} M_{(j,k)}(v_j^T v_k - \frac{\pi}{2} + \arccos(v_j^T v_k)) \tag{6}$$

The power series expansion of $\arccos(y)$ starts $\arccos(y) = \frac{\pi}{2} - y + \sum_{\ell \geq 3} c_\ell y^\ell$ where all the remaining coefficients $c_\ell$ are nonpositive, and converges on the entire interval $y \in [-1, 1]$. Thus Equation 6 equals

$$\sum_{j,k=1}^{n} \left( M_{(j,k)} \sum_{\ell \geq 3} c_\ell (v_j^T v_k)^\ell \right) \tag{7}$$

Since the matrix with $(j, k)$ entry $v_j^T v_k$ is positive semidefinite for any vectors $v_1, \ldots, v_n$, and since elementwise raising a positive semidefinite matrix to a positive integer power yields another positive semidefinite matrix, Equation 7 can be reexpressed as $\sum_{\ell \geq 3} \sum_{j,k=1}^{n} M_{(j,k)} N_{(j,k)}^{(\ell)}$ for some *negative* semidefinite matrices $N^{(\ell)}$, which is thus clearly less than or equal to 0, as desired. □

Combining the lower bounds and upper bounds of this section immediately yields Proposition 1.

## B   Proof of Theorem 3

For ease of exposition, we restate the Theorem 3:

**Theorem.** *Algorithm 1, given a description of the joint distribution of sample and target sets $(A_1, b_1), \ldots, (A_m, b_m)$, runs in polynomial time, and returns coefficients for a semilinear estimator whose expected squared error is within a $\frac{\pi}{2}$ factor of that of the best semilinear estimator. The value of the objective function achieved by $\widehat{V}$ is $m$ times the Proposition 1 SDP bound on the mean squared error of the best semilinear estimator.*

*Proof.* Proposition 1 describes a convex optimization problem to approximate to within a factor of $\frac{\pi}{2}$ the performance of an estimator specified by vectors $a_1, \ldots, a_m \in \mathbb{R}^n$. We thus consider optimizing Equation 2 over this choice (omitting the $\frac{1}{m}$ factor for convenience):

$$\min_{a_i : \{j : a_i(j) \neq 0\} \subseteq A_i} \max_{V \text{ psd}, V_{(j,j)} \leq 1} \sum_{i=1}^{m} \sum_{j,k=1}^{n} (a_i - b_i)_{(j)} (a_i - b_i)_{(k)} V_{(j,k)} \tag{8}$$

By Proposition 1, this minimum (if we can efficiently find it), will be within a factor of $\frac{\pi}{2}$ of the performance of the best semilinear estimator, and the vectors $a_1, \ldots, a_m$ that achieve this minimum will describe an estimator with this performance.

We proceed by invoking von Neumann's minimax theorem.

**Fact 2.** *Given a function $f(x, y)$ that is convex as a function of its first argument and concave as a function of its second argument, and given convex domains $X, Y$, at least one of which is bounded, then*

$$\min_{x \in X} \max_{y \in Y} f(x, y) = \max_{y \in Y} \min_{x \in X} f(x, y)$$

The condition that "at least one of $X, Y$ is bounded" is a relaxation of the original minimax theorem, shown sufficient by Sion [22].

We observe now that all the conditions of the minimax theorem are satisfied by the expression in Equation 8. As a function of $a_i$, the expression being optimized is the quadratic form with coefficients specified by the positive semidefinite matrix $V$; thus the expression is a convex function of $a_i$, and since such functions are summed over all $i \in \{1, \ldots, m\}$, the expression is a convex function of all the vectors $a_1 \ldots, a_m$. Since the expression is *linear* in $V$, it is thus also concave as a function of $V$. Finally, the domains of the vectors $a_1 \ldots, a_m$, along with the matrix $V$ are both convex, and, since a positive semidefinite matrix must have each entry bounded by the size of the largest diagonal entry, the condition that $V$ has diagonal entries bounded by 1 induces the same bound on the size of all entries of $V$.

Thus we invoke the minimax theorem to conclude that the value of Equation 8 is unchanged if we reverse the order of the min and the max:

$$\max_{V \text{ psd}, V_{(j,j)} \leq 1} \min_{a_i : \{j : a_i(j) \neq 0\} \subseteq A_i} \sum_{i=1}^{m} \sum_{j,k=1}^{n} (a_i - b_i)_{(j)} (a_i - b_i)_{(k)} V_{(j,k)} \tag{9}$$

Crucially, now, the inner minimization is simply a sum of positive semidefinite quadratic forms in each of the vectors $a_1, \ldots, a_m$. Reexpressing the inner sum in vector notation as $(a_i - b_i)^T V (a_i - b_i)$, the gradient of this quadratic form with respect to $a_i$ equals $2V(a_i - b_i)$. Thus, subject to the constraint that $a_i$ can only be nonzero on coordinates in $A_i$, if there exists a vector $a_i$ such that $V(a_i - b_i) = 0$ on coordinates $A_i$, then this $a_i$ attains the minimum; and otherwise the minimum is $-\infty$. The solution for $a_i$, restricted to the coordinates $A_i$ is thus $V_{(A_i, A_i)}^{-1} V_{(A_i)} b_i$ (or, when $V_{(A_i, A_i)}$ is singular, $V_{(A_i, A_i)}^{+} V_{(A_i)} b_i$ is the least-squares solution). Plugging this $a_i$ into the quadratic form yields $b_i^T (V - V_{(A_i)}^T V_{(A_i, A_i)}^{-1} V_{(A_i)}) b_i$ for the inner minimization of the $i^{\text{th}}$ term of the objective function. Finally, because of the setup of the minimax theorem, this expression must be a concave function of $V$, letting us conclude that Algorithm 1 can in fact conduct the optimization in polynomial time, as desired.

(As a side note, directly proving the above objective function is concave is a bit strange; it is a consequence of the fact that for positive definite $V$, and vectors $x$, the expression $x^T M^{-1} x$ is convex *as a function of both arguments*, implying it is convex even when both arguments are affine functions of the optimization variables.) $\qquad\square$

# C  Proof of Theorem 1: A Sample-Efficient Algorithm for Near-Optimal Semilinear Estimators

While Algorithm 1 takes as input the entire description $(A_1, B_1), \ldots, (A_m, B_m)$ of the joint sample/target distribution, such a description might be (1) unavailable in practice and/or (2) have support $m$ that is exponentially large. To address both cases, in this section we design an algorithm that achieves essentially the performance guarantees of Algorithm 1 (as given by Theorem 3), though relying only on *sampling* access to $P$. Algorithm 2 will run in time polynomial in $n$ and *independent* of the (possibly exponential) distribution description length $m$.

---

**Algorithm 2** Sampling algorithm to approximate the best semilinear estimator

---

**Input:** Accuracy parameter $\epsilon > 0$; $t$ random samples from the joint distribution of sample and target sets, $(A_{s_1}, b_{s_1}), \ldots, (A_{s_t}, b_{s_t})$, where each $A_i \subset \{1, \ldots, n\}$ is the set of sample set indices in the $i^{\text{th}}$ case and each $b_i$ is a vector with uniform values over the target set in the $i^{\text{th}}$ case as in Definition 4; and the actual instance to predict, specified by $(A, b)$ and the values $x_A$.

For an $n \times n$ matrix $V$ and a set $A_i \in \{1, \ldots, n\}$, let $V_{A_i}$ denote $V$ restricted to the rows in $A_i$, and let $V_{A_i, A_i}$ denote $V$ restricted to *both* rows and columns in $A_i$.

1. Compute the concave maximization

$$\widetilde{V} = \underset{V \succeq \epsilon,\, V_{j,j} \leq 1}{\arg\max} \sum_{i=1}^{t} b_{s_i}^T (V - V_{(A_{s_i})}^T V_{(A_{s_i}, A_{s_i})}^{-1} V_{(A_{s_i})}) b_{s_i} \tag{10}$$

2. Output the estimate $x_A \widetilde{V}_{(A,A)}^{-1} \widetilde{V}_{(A)} b$.

---

As compared with Algorithm 1, Algorithm 2 restricts the domain of optimization to matrices $V$ that have eigenvalues at least $\epsilon$, instead of at least 0 (which is a convex restriction). Crucially, instead of summing over all $m$ possible sample/target set possibilities, the optimization is over a small subset of size $t$, obtained by sampling. Finally, the output of this algorithm is phrased as a single estimate for the data in question (described to the algorithm via the triple $A, b, x_A$, as opposed to Algorithm 1, which returned the entire list of $m$ semilinear estimator coefficients). The following theorem, characterizing the performance of the above algorithm, immediately implies Theorem 1.

**Theorem 4.** *The mean squared error of the estimate output by Algorithm 2, over the randomness of the queried sample and target sets $(A, b)$, is within a multiplicative $\frac{\pi}{2}$ factor and an additive $6\epsilon$ factor of the performance of the optimum semilinear estimator, with probability $1 - e^{-t \cdot \epsilon^5 / poly(n)}$ over the sampled inputs $(A_{s_1}, b_{s_1}), \ldots, (A_{s_t}, b_{s_t})$. The probability of failure can thus be made exponentially small in $n$ by using $t = poly(n)/\epsilon^5$ samples, for a sufficiently large polynomial in $n$.*

We first prove three structural lemmas that characterize the optimization objective, and then put the pieces together making use of concentration bounds, applied over an $\epsilon$-net of matrices in the domain of the optimization.

**Lemma 1.** *For any valid $V$, the $i^{th}$ term in the sum of Equation 3—or equivalently Equation 9 or Equation 10—is between 0 and 1.*

*Proof.* From the derivation of Equation 3 in the proof of Theorem 3, the inner summation is equal to the inner minimization in Equation 9, which we analyze instead. Since the quadratic form specified by $V$ in Equation 9 is positive semidefinite, it thus always evaluates to a nonnegative number proving the first part of the claim.

Consider the inner minimum when all coefficients $a_i$ are identically 0. Since each $b_i$ is a nonnegative vector of sum 1, and thus since all entries of $V$ have magnitude at most 1 (because of the diagonal constraint, and the positive semidefinite constraint), we have $\sum_{j,k=1}^{n} b_{i(j)} b_{i(k)} V_{j,k} \leq 1$, as desired.

$\square$

**Lemma 2.** *The optimum objective value of the* max *in Equation 3 decreases by at most $\epsilon m$ if the domain of the maximization is further restricted so that $V$, instead of being positive semidefinite, must now have all eigenvalues at least $\epsilon$.*

*Proof.* From the derivation of Equation 3 in the proof of Theorem 3, the inner summation is equal to the inner minimization in Equation 9, which we analyze instead.

Letting $V$ be the optimal matrix in Equation 9 we instead consider the matrix $V_\epsilon = \epsilon I_n + (1 - \epsilon)V$ where $I_n$ is the $n \times n$ identity matrix. Since the objective is linear in $V$, when evaluated at $V_\epsilon$ it will have value $\epsilon$ times the objective value for $I_n$—which is nonnegative by Lemma 1—plus $(1 - \epsilon)$ times its optimal objective value at $V$—which is at most 1 by Lemma 1. Thus $V_\epsilon$ has objective value within $\epsilon$ of the optimum, as desired. □

**Lemma 3.** *For a fixed symmetric matrix $V$ whose eigenvalues are all at least $\epsilon$, the expression inside the sum of Equation 3, for any $i$, varies with respect to changing a coordinate of $V$ by at most $\left| \frac{d}{dV_{j,k}} b_i^T (V - V_{(A_i)}^T V_{(A_i, A_i)}^{-1} V_{(A_i)}) b_i \right| \leq \frac{1}{\epsilon^2} poly(n)$.*

*Proof.* Since $V$ has eigenvalues at least $\epsilon$, so does any (principal) submatrix $V_{(A_i, A_i)}$. Thus the inverse $V_{(A_i, A_i)}^{-1}$ has eigenvalues at most $\frac{1}{\epsilon}$, and thus the $L_2$ norm of any column of $V_{(A_i, A_i)}^{-1}$ is at most $\frac{1}{\epsilon}$. Since $\frac{d}{dV_{j,k}} V_{(A_i, A_i)}^{-1}$ equals negative the inner product of the columns (or rows) $j$ and $k$ of $V_{(A_i, A_i)}^{-1}$, this derivative is thus at most $\frac{1}{\epsilon^2}$. Applying the product rule can increase this by only a $poly(n)$ factor. □

We assemble these pieces to prove the performance of Algorithm 2.

*Proof of Theorem 4.* For any fixed $V$ in Equation 3, the average of the $m$ terms in the sum may be estimated as the empirical average of the $t$ terms we can compute from our randomly sampled inputs $(A_{s_1}, b_{s_1}), \ldots, (A_{s_t}, b_{s_t})$. Since, by Lemma 1, each term is between 0 and 1, the standard Chernoff/Hoeffding bounds imply that the empirical mean of $t$ random terms will be within $\epsilon$ of the true mean except with probability $e^{-2\epsilon^2 t}$.

Let $\epsilon' = \epsilon^3/poly(n)$ be a radius such that, by Lemma 3, any two matrices satisfying the constraints of the $\arg\max$ of Equation 10 that are within distance $\epsilon'$ of each other must yield values for each term in the sum, that are within $\epsilon$ of each other. Consider applying the concentration bounds of the previous paragraph to each $V$ in an $\epsilon'$-net of matrices satisfying the conditions of Equation 10—namely, positive definite with eigenvalues at least $\epsilon$, and all diagonal entries at most 1. Recall that an $\epsilon'$-net will have each matrix within distance $\epsilon'$ of one of the matrices in the net, and that the net will consist of $e^{poly(n)/\epsilon'}$ matrices. As we consider bounds up to $poly(n)$ factors, the choice of norm for the matrices does not matter, but for concreteness, consider the $\epsilon'$-net to be defined in the Frobenius norm. By the union bound, the Chernoff/Hoeffding bound of the previous paragraph applies for *every* $V$ in the $\epsilon'$-net except with probability $e^{-2\epsilon^2 t + poly(n)/\epsilon'}$, which is thus negligible when the number of samples is $t = poly(n)/\epsilon'\epsilon^2 = poly(n)/\epsilon^5$.

We thus show that the performance of the estimator described by the sampled $\widetilde{V}$ is close to the performance of the optimal semilinear estimator $\widehat{V}$ with eigenvalues at least $\epsilon$. Let $\widetilde{V}', \widehat{V}'$ respectively represent the nearest elements of the $\epsilon'$-net to $\widetilde{V}, \widehat{V}$ respectively. For ease of notation, we let $\widehat{f}(V)$ and $\widetilde{f}(V)$ respectively describe the functions of $V$ described by the average term in the sums of Equations 3 and 10 respectively. Thus we have

$$\widehat{f}(\widetilde{V}) \geq \widehat{f}(\widetilde{V}') - \epsilon \geq \widetilde{f}(\widetilde{V}') - 2\epsilon \geq \widetilde{f}(\widetilde{V}) - 3\epsilon \geq \widetilde{f}(\widehat{V}') - 3\epsilon \geq \widetilde{f}(\widehat{V}') - 4\epsilon \geq \widehat{f}(\widehat{V}) - 5\epsilon,$$

where the inequalities hold respectively because of (1) the $\epsilon'$-nearness of $\widetilde{V}, \widetilde{V}'$ combined with the derivative guarantee of Lemma 3 as applied to $\widehat{f}$; (2) the Chernoff/Hoeffding bound at the point $\widetilde{V}'$ of the $\epsilon'$-net; (3) the $\epsilon'$-nearness of $\widetilde{V}, \widetilde{V}'$ combined with the derivative guarantee of Lemma 3 as applied to $\widetilde{f}$; (4) the fact that $\widetilde{V}$ attains the maximum of $\widetilde{f}$; (5) the Chernoff/Hoeffding bound at the point $\widetilde{V}'$ of the $\epsilon'$-net; and (6) the $\epsilon'$-nearness of $\widehat{V}, \widehat{V}'$ combined with the derivative guarantee of Lemma 3 as applied to $\widehat{f}$.

Thus, the algorithm described by $\widetilde{V}$ has true performance within $5\epsilon$ of the optimal under the eigenvalue constraint, achieved by $\widehat{V}$. By Lemma 2, $\widehat{V}$ itself is within $\epsilon$ of the true optimal performance of Equation 3, which in turn is within a factor of $\frac{\pi}{2}$ of that of the best semilinear estimator, as desired. $\quad\square$

## D  Suboptimality of Semilinear Schemes (Fact 1)

Via a computer-aided brute-force search over small examples, we found a distribution, $P$, for which the best semilinear algorithm had larger worst-case expected error than the best arbitrary scheme:

**Example 4.** *Let $n = 4$. Consider the distribution over sampling from the population $\{1, 2, 3, 4\}$ that assigns a 0.3 probability to the following pairs of sample/target sets $(\{1, 3\}, \{2, 4\})$, $(\{2, 4\}, \{1, 3\})$, $(\{3, 4\}, \{1, 2\})$ and a 0.05 probability to $(\{1, 3, 4\}, \{2\})$ and $(\{2, 3, 4\}, \{1\})$. The optimal semilinear scheme achieves worst-case expected squared error 0.6652, compared to 0.6627 for the optimal unconstrained scheme. Hence even for sample/target set distributions over $n = 4$ datapoints, semilinear schemes are not always worst-case optimal.*

## E  Proof of Theorem 2: Linear Regression Setting

We prove Theorem 2 here, which for clarity we restate and reintroduce slightly more formally.

We consider the following natural extension of our results to the setting of $d$-dimensional linear regression. Our regression results follow from a transparent application of the main results, Theorem 1 or Theorem 3, demonstrating the flexibility and scope of our approach. We emphasize that many variants of this model are interesting, and that a more specialized analysis may yield stronger bounds than what we show here.

**Definition 9.** *Given a sample set $A$ and a target set $B$, where for each $i \in A$ we are given a pair $(x_i, y_i)$ with the independent variable $x_i \in [-1, 1]^d$ and the dependent variable $y_i \in [-1, 1]$, the goal is to recover a coefficient vector $\beta$ that minimizes the mean squared error on the target set, $\mathbb{E}_{i \sim B}[(x_i^\top \beta - y_i)^2]$, when additionally given access to the joint distribution $P$ from which $(A, B)$ are drawn. The least-squares coefficients are defined, as is standard, as $\beta = (E_{i \sim B}[x_i x_i^\top])^{-1} E_{i \sim B}[x_i y_i]$. As we do not have access to the target set data $(x_i, y_i)$ for $i \in B$, we instead must estimate it: depending on whether distribution $P$ is given explicitly, or via sample access, use Theorems 1 or 3 respectively to estimate (in terms of $P$ and the sample data $(x_i, y_i)$ for $i \in A$) each of the $d^2$ entries in the matrix $Q = E_{i \in B}[x_i x_i^\top]$ and the $d$ entries in the vector $u = E_{i \in B}[x_i y_i]$, all of which are in $[-1, 1]$. Our estimated coefficients are then $\hat{\beta} = \widehat{Q}^{-1} \hat{u}$.*

**Theorem.** *Given a regression problem as in Definition 9, where the distribution $P$ is specified either via sampling access or explicitly, and let $\alpha_P$ be the mean squared error guaranteed by Theorem 1 or 3 respectively, for estimating the mean of a scalar (which depends only on $P$). Then, letting $\sigma_d$ be the smallest singular value of the (uncentered covariance) matrix $Q$, for any $\delta > 0$, the algorithm of Definition 9 will return an estimate $\hat{\beta}$ of the least-squares regression coefficients $\beta$ such that with probability at least $1 - \delta$ we have $||\beta - \hat{\beta}|| \leq 3 \frac{\sqrt{\alpha_P d^3 / \delta}}{\sigma_d^2}$, provided this expression is at most $0.08$; in the case that the independent variables $x_i$ are known for the target set (and only the $y_i$'s are unknown) then except with $\delta$ probability, we have $||\beta - \hat{\beta}|| \leq \frac{\sqrt{\alpha_P d / \delta}}{\sigma_d}$.*

*Proof.* Recall we are in the following regression setting: data consists of pairs $(x_i, y_i)$ where $x_i \in [-1, 1]^d$ and $y_i \in [-1, 1]$; indices $i \in [n]$ are drawn for the input sample $A$ and target set $B$ from a joint distribution, $(A, B) \sim P$. The goal, given the training data and a description of $P$ (or sample access to $P$), is to compute linear coefficients $\beta \in \mathbb{R}^d$, such that the mean squared error over target indices $i \in B$ is as small as possible, namely, to minimize $\mathbb{E}_{i \sim B}[(x_i^\top \beta - y_i)^2]$. Our results will be parameterized in terms of $\alpha_P$, the mean squared estimation accuracy that Theorem 1 or Theorem 3 affords us on distribution $P$ (in the scalar mean estimation setting).

As is standard in least squares regression, note that the expression we are minimizing, $\mathbb{E}_{i \sim B}[(x_i^\top \beta - y_i)^2]$, is positive semidefinite as a function of $\beta$, and thus is minimized when its gradient with respect to $\beta$ equals 0. Hence we solve $\mathbb{E}_{i \sim B}[x_i(x_i^\top \beta - y_i)] = 0$, or equivalently, $\mathbb{E}_{i \sim B}[x_i x_i^\top]\beta = \mathbb{E}_{i \sim B}[x_i y_i]$, which has solution $\beta = (\mathbb{E}_{i \sim B}[x_i x_i^\top])^{-1} \mathbb{E}_{i \sim B}[x_i y_i]$. Now, for any $i$, each of the $d^2$

entries of the matrix $x_i x_i^\top$ is in $[-1, 1]$, and thus its average value over $i$ in the target set $B$ can be estimated to within mean squared error $\alpha_P$ based on its values for $i$ in the sample set $A$, from Theorem 1 or 3; the same holds for each of the $d$ entries in the vector $x_i y_i$.

Let $Q = \mathbb{E}_{i \sim B}[x_i x_i^\top]$, and let $Q + F$ (referred to as $\widehat{Q}$ in definition 9) be the random variable representing the estimate of $Q$ given by Theorem 1 or 3, where the square of each entry of $F$ has expectation at most $\alpha_P$; hence by Markov's inequality, for any $\delta > 0$, with probability at least $1 - \delta$ the matrix $F$ has Frobenius norm at most $\sqrt{\alpha_P d^2/\delta}$. Correspondingly, let $u = \mathbb{E}_{i \sim B}[x_i y_i]$, and let $u + g$ be the random variable representing the estimate of $u$ returned by our algorithm, where, except with probability $\delta$, the vector $g$ has length at most $\sqrt{\alpha_P d/\delta}$. Taking the union bound, we have that except with probability $2\delta$, the bounds on both $F$ and $g$ hold, and we analyze this case below.

As described above, the optimal linear coefficients are given by $\beta = Q^{-1} u$, and meanwhile our estimate is $\hat{\beta} = (Q + F)^{-1}(u + g)$. We bound the discrepancy $||\beta - \hat{\beta}||$ via the triangle inequality, first bounding the change induced by adding $g$, and then bounding the change from adding $F$:

$$||\beta - \hat{\beta}|| = ||Q^{-1}u - (Q+F)^{-1}(u+g)|| \le ||Q^{-1}u - Q^{-1}(u+g)|| + ||Q^{-1}(u+g) - (Q+F)^{-1}(u+g)||$$

The first term on the right hand side equals $||Q^{-1}g||$, which we bound as the product of the length of $g$ and the largest singular value of $Q^{-1}$, which is the inverse of the smallest singular value of $Q$, which we have denoted $\sigma_d$. Thus we have $||Q^{-1}g|| \le \frac{1}{\sigma_d}\sqrt{\alpha_P d/\delta}$.

Bounding the second term is slightly more involved. The goal is to bound $||[Q^{-1} - (Q+F)^{-1}](u+g)||$. We first bound the largest singular value of the matrix $Q^{-1} - (Q+F)^{-1}$. For any $\lambda \in [0, 1]$, consider interpolating between $Q$ and $Q + F$ to get $Q + \lambda F$. Let $s_\lambda$ be the smallest singular value of this matrix, and let $v_\lambda$ be the corresponding singular vector, with $||v|| = 1$. Since $Q$ has smallest singular value $\sigma_d$, we have $||Qv|| \ge \sigma_d$; since $F$ has Frobenius norm at most $\sqrt{\alpha_P d^2/\delta}$ and the Frobenius norm bounds the largest singular value, we have $||Fv|| \le \sqrt{\alpha_P d^2/\delta}$, and by the triangle inequality, the difference of these two expressions is a lower bound on $s$: $s = ||(Q + F)v|| \ge \sigma_d - \sqrt{\alpha_P d^2/\delta}$.

Consider the matrix $(Q + \lambda F)^{-1}$ as we linearly move $\lambda$ from 0 to 1. The derivative with respect to $\lambda$ of this matrix inverse is $-(Q^{-1} + \lambda F)F(Q + \lambda F)^{-1}$, which thus has largest singular value at most the product of our bounds on the singular values for the 3 terms:

$$\frac{\sqrt{\alpha_P d^2/\delta}}{(\sigma_d - \sqrt{\alpha_P d^2/\delta})^2} \tag{11}$$

Since $||u + g|| \le \sqrt{d} + \sqrt{\alpha_P d/\delta}$ from summing bounds on $||u||$ and $||g||$, we multiply this by Equation 11—bounding the amount the matrix $(Q + \lambda F)^{-1}$ changes as we interpolate from $Q$ to $Q + F$—to get our total bound for the second triangle inequality term. Adding this to the bound on the first term, we have

$$||\beta - \hat{\beta}|| = ||(Q+F)^{-1}(u+g) - Q^{-1}u|| \le \frac{1}{\sigma_d}\sqrt{\alpha_P d/\delta} + (\sqrt{d} + \sqrt{\alpha_P d/\delta})\frac{\sqrt{\alpha_P d^2/\delta}}{(\sigma_d - \sqrt{\alpha_P d^2/\delta})^2} \tag{12}$$

To simplify this bound, consider the case that $\frac{\sqrt{\alpha_P d^3/\delta}}{\sigma_d^2} \le c$.

Since matrix $Q$ has entries in $[-1, 1]$, its singular values are at most $\sqrt{d}$, and thus $\frac{\sqrt{d}}{\sigma_d} \ge 1$, yielding $\frac{\sqrt{\alpha_P d^2/\delta}}{\sigma_d} \le c$, and implying that if we replace the denominator $(\sigma_d - \sqrt{\alpha_P d^2/\delta})^2$ in Equation 12 by simply $\sigma_d^2$ then the expression will decrease by at most a factor of $(1 - c)^2$. Similarly, in Equation 12 the parenthetical term $(\sqrt{d} + \sqrt{\alpha_P d/\delta})$ has second part bounded by $\sqrt{\alpha_P d/\delta} \le c\frac{\sigma_d}{\sqrt{d}} \le c$ and thus replacing $(\sqrt{d} + \sqrt{\alpha_P d/\delta})$ by simply $\sqrt{d}$ will decrease the term by at most a factor of $\frac{1}{1+c}$. Thus the right hand side of Equation 12 is the sum of two terms, $\frac{\sqrt{\alpha_P d/\delta}}{\sigma_d}$, and a term $\frac{\sqrt{\alpha_P d^3/\delta}}{\sigma_d^2}$ times a factor between 1 and $\frac{1+c}{(1-c)^2}$; the first term is clearly at most the second term since $\sigma_d \le \sqrt{d} \le d^2$, so thus we have $||\beta - \hat{\beta}|| \le (1 + \frac{1+c}{(1-c)^2})\frac{\sqrt{\alpha_P d^3/\delta}}{\sigma_d^2}$. Substituting $2\delta \to \delta$ so that the overall probability

of failure becomes $\delta$, the proportionality constant becomes $\sqrt{2}(1 + \frac{1+c}{(1-c)^2})$, which for $c \leq 0.0378$ yields a constant of 3, as desired. Thus, in the context of the theorem, substituting in $\frac{1}{2}\delta$ for $\delta$, when $3\frac{\sqrt{\alpha_P d^3/(\delta/2)}}{\sigma_d^2} \leq 0.08$, we will have $c \leq \frac{0.08\sqrt{2}}{3} < 0.378$ as desired.

In the simpler case where the independent variables $x_i$ are known, and hence $Q$ does not need to be estimated, our probability of failure is $\delta$ instead of $2\delta$, and only the first term from Equation 12 appears, immediately yielding the other part of the theorem.

$\square$