[Reviews · NeurIPS 2020]

Review 1

Summary and Contributions: This paper introduces a model for statistics for settings where standard assumptions like "data is i.i.d. from a distribution in a known class" may be (badly) violated. The authors give an algorithm for estimation in this model and prove that it incurs error comparable to the optimal algorithm in a natural but restricted class of estimators. The authors provide convincing numerical experiments demonstrating that their estimation algorithm outperforms standard strategies.

Strengths: Let me first summarize the paper somewhat more thoroughly. The model introduced by the authors is as follows. There are $n$ data points $\{x_1,\ldots,x_n\} \in [-1,-1]$; these may be arbitrary and unknown. (In particular they do not have to be Gaussian distributed, or anything like that.) There is a (known, or accessible via samples) distribution $P$ on pairs $(A,B)$ where $A,B \subseteq [n]$. The set $A$ is the set of "revealed" data points -- samples an estimation algorithm has access to. The goal is to estimate some statistic of the data points in $B$, given $A,B$ and the values $x_i$ for $i \in A$. For this review I will just discuss estimating the mean of the values in $B$, although the approach in the paper extends to estimating variance or higher moments as well as to regression problems. Notice that this model allows for situations where the sampling procedure is far from i.i.d. sampling -- in particular, the value $x_i$ of the $i$-th point may have a large effect on whether $i$ is included in $A$ or $B$. The authors study a natural class of estimation strategies for such problems, which they call "semilinear". These are estimators which always output a linear function of the observed values $x$, but where the coefficients of that linear function may depend *arbitrarily* on the realization of $A$ and $B$. This captures standard schemes like importance sampling, as well as a number of interesting additional settings described in the paper. The main result is an efficient algorithm which takes the distribution $P$ (in the form of polynomially-many samples) and outputs an approximately-optimal semilinear estimation strategy (i.e. a rule mapping $A,B$ to a linear estimator, i.e. a weighting scheme for the samples in $A$). The "approximate" in "approximately-optimal" means that the expected in estimating $B$ is at most a constant factor greater than what would be incurred by the optimal semilinear estimator given worst-case data. The authors also extends this result to the linear regression setting. The paper also includes some very interesting numerical experiments in natural settings such as "snowball sampling" where samples are collected via a social network, and demonstrate that their estimator significantly out-performs the sample mean. I think this paper should be of great interest to the NeurIPS community. It checks all the boxes for a great NeurIPS paper: -- focus on a critical problem (how to do statistics in the face of correlated sampling and adversarial data?) -- introduce a new model capturing important existing settings and suggesting new ones -- nice theory result giving an algorithm with strong provable guarantees -- convincing numerical experiments. -- directions for future work

Weaknesses: The paper has few weaknesses. It would be nice if the experiments were for some more realistic settings. The algorithm suggested is probably not practical for more than 50-100 data points, because it requires solving a semidefinite program -- it would be interesting to obtain an algorithm with similar guarantees which scales to larger data sets.

Correctness: I did not check all the proofs line by line, but based on the sketches provided by the authors I think the results are likely to be correct.

Clarity: The paper is well written.

Relation to Prior Work: Is adequately discussed.

Reproducibility: Yes

Additional Feedback: It would be nice to offer some additional argument for lines 270-272.


Review 2

Summary and Contributions: The paper introduces a new framework for statistical estimation, where we assume knowledge of a distribution which describes the sampling process of the sample set (in machine learning terms, a training set) and the target set, but allows arbitrary values for collected samples (subject to some natural boundedness conditions). Within this general setup the paper provides an efficient algorithm for target mean estimation and its guarantees via a connection to the Grothendieck problem.

Strengths: In my opinion this is a strong submission, especially due to its conceptual novelty. This work arguably falls under the umbrella of robust statistics, however the typical formalization of robust statistics involves some true underlying target distribution, and then the training distribution is obtained as a corrupted version of this target distribution. Here, on the other hand, the assumed distribution is only over the collected samples (in a sense, it’s like assuming a distribution over which samples will be truncated), and then the data values can be arbitrary. The current paper focuses on a relatively simple setting of semilinear schemes, however it’s a very good stab at analyzing this new “randomly collected, worst case data” formalism. I think this submission could inspire a lot of follow-up work.

Weaknesses: I have a couple of stylistic comments. When describing the linear regression setup, I’d add a half-sentence remark clarifying that you’re assuming the model is identifiable (otherwise beta_B is not well-defined). Preceding Definition 2, I was a bit confused by the wording “approximated to arbitrary accuracy”. Given that everything is finite, there is no “approximation”, but you can come up with the appropriate uniform distribution which exactly matches P. Finally, I’m not a fan of the term “sample set”, since technically B is also a sample set (i.e. set of samples). It’s possible that this is some standard terminology though, in which case it’s probably best to stick to whatever is standard. Also, in the future it would probably be good to see what one can say under different constraints of the samples; assuming point-wise boundedness is good when we envision something like i.i.d. samples. However, many natural applications where the samples from A are "coupled" imply, say, an l1 or l2 constraint on the sample set, rather than l-infinity.

Correctness: Yes, the claims seem correct.

Clarity: The paper is very clearly written, I don't have any suggestions regarding this.

Relation to Prior Work: Yes, the Related Work section is thorough.

Reproducibility: Yes

Additional Feedback:


Review 3

Summary and Contributions: This paper introduces a new model for statistical estimation, which does not assume the data come from a specific distribution. In contrast, the model is based on an assumption on how data are collected. More precisely, for any set of population elements x_1,...,x_n, the estimator draws tuples of sets of indices in [n], (A,B) from a distribution P, where A is the sample set and B is the target set. The estimator only sees the values of x_i, i\in A and aims to give an estimate of the statistic on B. Its performance is measured by the worst case (in terms of population elements x_1,...,x_n) expected error over P. The authors give an algorithm for estimating the target mean (both for the case that P is known and when it is accessed via sampling), and prove its approximation guarantee with respect to the best estimator, for this class of estimators (semi-linear estimators). They also apply their results to the setting of linear regression, giving an estimator for the least-squares regression model of the target set. Finally, the authors demonstrate the better performance of their mean estimator for various sampling processes.

Strengths: -This paper is novel in the sense that it introduces a new model for statistical estimation. This model is based on the assumptions on the sampling process rather than distributional assumptions over the data themselves and it would be interesting if it could yield insights and bounds for cases where distributional assumptions fail (e.g. the important case where data are not drawn iid). The model seems meaningful as it encompasses some of the interesting sampling processes (e.g. snowball sampling, chronological sample structure). -There are interesting mathematical connections between the class of semi-linear estimators and semi-definite programming (that lead to an elegant proof for the approximation factor of the mean estimator).

Weaknesses: -The results are a bit restrictive as they are limited to the class of semi-linear estimators for linear statistics (i.e., sums of functionals of each datapoint in the set). The authors also empirically find that exists a distribution for which the best semi-linear estimator is sub-optimal. However, their structure allows for the clean analysis of their performance (via convex semi-definite optimization). -If we consider n to be the size of the whole population, then the number of samples (tuples of sample/target sets) needed by the estimator, which is poly(n), is quite large. So overall, the results intended to demonstrate the usefulness of the model are not particularly strong. However, this is not necessarily a requirement for a paper introducing a new model.

Correctness: The claims are fully proven and the set-up and results of the experiments are clearly explained.

Clarity: The paper is very well written. I particularly liked the set of three (!) examples aiming to explain why the model is meaningful.

Relation to Prior Work: Prior work mostly focused on removing the iid assumption, but not on removing the distributional assumption completely and was sufficiently discussed.

Reproducibility: Yes

Additional Feedback: Again, I found the presentation of the paper very good! Besides some low-level comments, my main suggestion would be to try to compare this model with the common case where we have a distribution D on the population and aim to minimize the statistic over D. Are there cases where a bound on the benchmark you propose implies a bound on the error of the estimator of expected value of the statistic over D? A reduction or settings where comparison is easy to do would give more intuition about how strict the model is (and set the tone on what bounds we can expect from it). Similarly, is there an interpretation of your mean estimation result in this setting? -Line 211: exp-> \exp -Line 292: provides -> provide -Eq. (6)+(7): j_k -> v_k ============================== Thank you for your response. I increased my score a bit, leaning more on the novelty of the model and the potential for future work. I apologize that my suggestion was not clear. Let me elaborate: As a reader, I found it hard to estimate how "strict" your new model is. And this, in turn, makes it hard to guess whether we can have interesting results using this model in the future. By "strict", I mean whether this benchmark (of the optimal worst-case x_i - expected over P error) is high to begin with, for interesting problems. If that were the case, it might be better to focus our research on case-by-case approaches/models that involve more assumptions (e.g. the data come from a distribution D but the sample A only includes truncated samples, and so on). I feel that this concern could be addressed if we considered a problem, where this optimal benchmark you propose is not too far from the optimal error of a more standard model (and I know that if that were the standard model of drawing the sample i.i.d. from a distribution, then this is too much to ask). I don't have a particular suggestion for a problem and model to compare to. If you can think of such an example to help the reader understand what they can expect from your new model, then that's great. If not, that's also okay.


Review 4

Summary and Contributions: The paper proposes a framework to study a particular kind of subsampling process, where from a finite set two sets are subsampled. The analysis and techniques assume knowledge of the subsampling process but no knowledge of the data distribution is required. The results primarily apply to simple semi-linear estimators which, the authors motivate as being relevant.

Strengths: 1) Novel setting 2) Well written 3) Analysis and principled algorithm provided

Weaknesses: 1) The main assumption of access to P may not hold in many real settings 2) Results mainly apply to scalar sets 3) Results apply to the restricted class of semi-linear estimators, and is not clear how more complex functions could be accomodated

Correctness: Yes, as far as I could verify

Clarity: Yes

Relation to Prior Work: Yes

Reproducibility: Yes

Additional Feedback: The paper proposes a framework to study a particular kind of subsampling process, where from a finite set two sets are subsampled. The analysis and techniques assume knowledge of the subsampling process but no knowledge of the data distribution is required. The results primarily apply to simple semi-linear estimators which, the authors motivate as being relevant. I think this is an interesting problem setup where the analysis is based on the subsampling process rather than distribution of the data. The analysis and the algorithm is interesting although primarily restricted to semi-linear models with scalar sets. My main concerns are regarding the realisticness of the setup. The setup assumes subselection from a finite set, which may not be the case for many real machine learning problems. Typically, a bound on n is not known or n is large where the set B is much larger than the set A with it being data collected over time by a deployed system. Another issue is P or the sampling bias is hard to ascertain in many practical circumstances. In some sense, its easier to obtain an estimate for the actual data distribution that the sampling bias as data can come from multiple sources and based on unknown data collection schemes. ---------------------- Post Rebuttal The results for semi-linear models are interesting but some comments on extension to more realistic models would be good for a general NeurIPS reader to have. All in all though given the novelty of the idea I am raising my score.

[Author Response · NeurIPS 2020]

Thanks for your comments and helpful suggestions! We begin with responses that might interest multiple reviewers.

REVIEWER 4: Framework requires knowledge of $P$, but in many practical settings it is difficult/impossible to know $P$, or control/design $P$.

We believe that the difficulty of knowing $P$ in some practical settings makes our work *more* applicable, not less, in the following sense. Suppose someone conducts a poll or survey of students and then makes a claim like "I estimate the overall student body mean to be 74 based on the empirical mean of the respondents". No one truly knows the underlying sampling distribution, $P$ (different students have different response probabilities, there might be some independence, but also likely correlations in participation between friends, etc.). Using our framework and algorithm, you could design a variety of plausible sampling distributions, $P$, and then evaluate the worst-case expected error of the empirical mean with respect to these $P$'s, which would provide compelling evidence for whether or not to believe the claim about the student body mean being 74. [Recall Prop 1 in Sec 2.1 that argues that we can evaluate the worst-case-expected-error of a given/fixed (semi-linear) estimator such as the empirical mean, with respect to these "plausible" $P$'s.]

In this sense, our framework and algorithm can be used to rigorously evaluate the stability of an estimator (in the above example, the empirical mean) with respect to various "plausible" $P$'s. We think this is a useful alternative to more standard approaches that evaluate the estimator with respect to other kinds of assumptions on the data. We plan to add a discussion of this to the paper and thank the reviewer for eliciting this alternate use case for our framework.

REVIEWER 2: Extension to $\ell_1$ or $\ell_2$ constraints instead of $\ell_\infty$.

Good question. Our original motivation was for settings such as surveys/polling (e.g. COVID testing) where data values are binary or otherwise bounded in magnitude, and where $\ell_\infty$ is the natural constraint. We agree that investigating our model with respect to other constraints seems natural and worthwhile. We have done some preliminary work investigating the $\ell_2$ case (not included in our submission): since the geometry of the $\ell_2$ norm is so well behaved, there is potential for efficient algorithms that might even improve upon the $\pi/2$ approximation factor and surmount several of the more structural hurdles we encountered when investigating fully general (non semi-linear) estimators. We haven't fleshed out any details, though we agree this is a very interesting direction.

REVIEWER 4: Underlying domain is assumed to be finite (ie size $n$).

Our framework and definition of worst-case expected error apply, without modification, to infinite sets of underlying datapoints. Additionally, it seems like an extension of our algorithm might be adapted to such settings. To briefly sketch this, suppose $P$ corresponds to a joint distribution over an infinite (or continuous) domain indexing potential elements, and that each sample/target set in the support of $P$ has size at most $k$. Very roughly, one can (1) draw many (i.e. $poly(k)$) sample/target sets from $P$, let $Z$ denote the union of these sampled sets (together with the $\leq 2k$ elements of $A$, $B$–the actual sample/target sets) and then (2) let $P'$ denote the restriction of $P$ to those sample/target sets that have non-empty intersections with $Z$, and (3) run our algorithm on this (now finite) set $Z$ and distribution $P'$. The one delicate step (that would take some space to fully describe) is that one must adjust $P'$ to account for the possibility that the measure in $P$ that intersects $Z$ in 1 point, might be (infinitely) larger than the measure intersecting in 2 points, etc.

This extension of our algorithm to infinite/continuously indexed data is certainly not trivial, and we haven't fleshed out a formal proof of this. It does seem quite interesting—thanks for pointing out this direction.

REVIEWER 4: "Results mainly apply to scalar sets"

We mainly focus on scalar sets, as that seems like the natural starting point for explaining/exploring this framework. Still, as our linear regression results (Thm 2) illustrate, the framework naturally extends to non-scalar settings, and one can obtain interesting results in these non-scalar settings. We imagine that future work will likely explore a number of different non-scalar settings within the framework we propose.

REVIEWER 3: "try to compare this model with the common case where we have a distribution D on the population and aim to minimize the statistic over D. Are there cases where a bound on the benchmark you propose implies a bound on the error of the estimator of expected value of the statistic over D?"

We aren't sure we understand this question. Our interpretation of your question is the following: Suppose we have a distribution $D$ (ie over all images) and the ultimate goal is to train a model to classify, say, cat images, that has small expected loss wrt $D$. Given a model, how do we evaluate the expected loss over $D$? If we can sample from $D$, then great. If we can't sample from $D$, but instead can generate some (possibly dependent) set of samples $S$, drawn from a joint distribution $D'$ over sets of $k = |S|$ samples, then we *could* use our framework to figure out how to estimate the expected loss over $D$ based on the values of the loss on set $S$. [One natural example of such a $D'$ might correspond to taking a sequence of $k$ samples from a Markov Process whose stationary distribution is $D$...] We not sure if there are any 'standard' instances of such settings for which there are clean results to which we could compare with. [If this doesn't address your question, please do clarify your question in your review...]

[Meta-Review · NeurIPS 2020]

The paper proposes a new framework statistical analysis, which pushes randomness to the sampling strategy, but allows the actually data points to be arbitrary. Apart from introducing this model, the paper also studies a mean estimation problem in this framework and provide some approximation guarantees for a proposed estimator. The reviewers (and I) believe this model is very interesting and likely to spur much future work. In addition to the model, the actual results of the paper are also quite strong. An oral presentation would provide the authors with a high-profile forum to share the results with the community.